# Detection of Arrhythmias Using Smartwatches—A Systematic Literature Review

**DOI:** 10.3390/healthcare12090892

**Published:** 2024-04-25

**Authors:** Bence Bogár, Dániel Pető, Dávid Sipos, Gábor Füredi, Antónia Keszthelyi, József Betlehem, Attila András Pandur

**Affiliations:** 1Department of Oxyology and Emergency Care, Pedagogy of Health and Nursing Sciences, Institute of Emergency Care, Faculty of Health Sciences, University of Pécs, 7624 Pécs, Hungary; daniel.peto@etk.pte.hu (D.P.); gabor.furedi@etk.pte.hu (G.F.); betlehem@etk.pte.hu (J.B.); attila.pandur@etk.pte.hu (A.A.P.); 2Department of Medical Imaging, Faculty of Health Sciences, University of Pécs, 7400 Kaposvár, Hungary; david.sipos@etk.pte.hu; 3Human Patient Simulation Center for Health Sciences, Faculty of Health Sciences, University of Pécs, 7624 Pécs, Hungary; keszthelyi.antonia@pte.hu

**Keywords:** arrhythmia, atrial fibrillation, smartwatch

## Abstract

Smartwatches represent one of the most widely adopted technological innovations among wearable devices. Their evolution has equipped them with an increasing array of features, including the capability to record an electrocardiogram. This functionality allows users to detect potential arrhythmias, enabling prompt intervention or monitoring of existing arrhythmias, such as atrial fibrillation. In our research, we aimed to compile case reports, case series, and cohort studies from the Web of Science, PubMed, Scopus, and Embase databases published until 1 August 2023. The search employed keywords such as “Smart Watch”, “Apple Watch”, “Samsung Gear”, “Samsung Galaxy Watch”, “Google Pixel Watch”, “Fitbit”, “Huawei Watch”, “Withings”, “Garmin”, “Atrial Fibrillation”, “Supraventricular Tachycardia”, “Cardiac Arrhythmia”, “Ventricular Tachycardia”, “Atrioventricular Nodal Reentrant Tachycardia”, “Atrioventricular Reentrant Tachycardia”, “Heart Block”, “Atrial Flutter”, “Ectopic Atrial Tachycardia”, and “Bradyarrhythmia.” We obtained a total of 758 results, from which we selected 57 articles, including 33 case reports and case series, as well as 24 cohort studies. Most of the scientific works focused on atrial fibrillation, which is often detected using Apple Watches. Nevertheless, we also included articles investigating arrhythmias with the potential for circulatory collapse without immediate intervention. This systematic literature review provides a comprehensive overview of the current state of research on arrhythmia detection using smartwatches. Through further research, it may be possible to develop a care protocol that integrates arrhythmias recorded by smartwatches, allowing for timely access to appropriate medical care for patients. Additionally, continuous monitoring of existing arrhythmias using smartwatches could facilitate the assessment of the effectiveness of prescribed therapies.

## 1. Introduction

Among the challenges encountered in emergency departments, cardiovascular disorders stand out as the most common and severe conditions, contributing significantly to global morbidity and mortality [1]. Globally, cardiovascular diseases are recognized as a leading cause of death, accounting for an estimated 17.9 million lives annually, constituting approximately 45% of all deaths [2,3]. In Europe, over 1.4 million premature deaths occur annually due to cardiovascular diseases in individuals under the age of 75 [3]. Common cardiovascular conditions include myocardial infarction, stroke, heart failure, cardiac arrhythmias, and heart valve issues [4].

Among cardiac arrhythmias, atrial fibrillation (AF) is the most prevalent, affecting 8.8 million individuals aged 55 and older in Europe in 2010. Projections indicate that this number will increase to more than double to 17.9 million by 2060 [5]. However, various other arrhythmias may develop, posing potential threats to patients’ lives, including sinus tachycardia, atrial flutter, supraventricular tachycardia, ventricular fibrillation, ventricular tachycardia, sinus arrest, sick sinus syndrome, or atrioventricular blocks [6]. Arrhythmias are associated with 15–20% of all deaths, particularly sudden cardiac death, emphasizing the need for heightened attention to these conditions [7].

Fortunately, advancements in technology have introduced wearable smart devices, such as smartwatches, capable of assisting in the detection and management of cardiac arrhythmias and various health conditions [8,9]. Wearable smart devices have become one of the fastest-growing sectors in the technology industry, with major tech companies like Apple (Apple Watch), Google (Fitbit), and Samsung (Galaxy) developing smartwatches capable of monitoring biometric data, including heart rhythm, pulse rate, oxygen saturation, blood pressure, and sleep pattern [10]. Some devices utilizing photoplethysmography (PPG) can register patients’ electrocardiography (ECG) within a 30-s interval, playing a crucial role in monitoring AF [11,12,13].

Beyond detecting AF, smartwatches can prove valuable in identifying other ECG abnormalities, such as bradyarrhythmias, tachyarrhythmias, or deviations indicative of ischemia [14]. With their current capabilities, smartwatches can provide excellent support for healthcare professionals in recognizing and managing various ECG abnormalities [8]. In our current research, we systematically aim to compile literature that specifically focuses on the registration of ECG abnormalities via smartwatches, particularly those relating to arrhythmias.

## 2. Methods

Our systematic literature review gathered available case reports, case series, and cohort studies. The literature review was conducted following the 2020 PRISMA (Preferred Reporting Items for Systematic Reviews and Meta-Analyses) guidelines, utilizing the PRISMA 2020 Checklist for the article’s preparation [15].

### 2.1. Procedure for Literature Search

The search encompassed four databases: Web of Science, PubMed, Scopus, and Embase. Specifically, we sought articles and research focusing on the detection or monitoring of arrhythmias and other ECG abnormalities utilizing smartwatches. The systematic literature review included scientific works published from 1 January 2019 to 1 August 2023.

During the research, we utilized the following keywords: “smart watch” OR “smartwatch” OR “smart watches” OR “smartwatches” AND “Apple Watch” OR “Samsung Gear” OR “Samsung Galaxy Watch” OR “Google Pixel Watch” OR “Fitbit” OR “Huawei Watch” OR “Withings” OR “Garmin” AND “Atrial Fibrillation” OR “Supraventricular Tachycardia” OR “Cardiac Arrhythmia” OR “Ventricular Tachycardia” OR “Atrioventricular Nodal Reentrant Tachycardia” OR “Atrioventricular Reentrant Tachycardia” OR “Heart Block” OR “Atrial Flutter” OR “Ectopic Atrial Tachycardia” OR “Bradyarrhythmia”. We experimented with various combinations of keywords and utilized Boolean operators to refine the search results. These searches were complemented with keywords and MeSH terms to broaden the scope of the findings. Additionally, we examined the bibliography of the selected literature to identify further relevant articles for inclusion.

Initially, we filtered articles based on titles and abstracts. Subsequently, we selected scientific works that were written in English or German and identified arrhythmias or other ECG abnormalities using smartwatches. We excluded conference abstracts, editorials, letters, guidelines, literature reviews, and meta-analyses from this systematic literature review. Studies issued by tech companies that manufacture smartwatches were also excluded.

### 2.2. Quality Assessment

Using the Newcastle-Ottawa Scale, we assessed the methodological quality and appropriateness of all case reports, case series, and cohort studies [16,17].

### 2.3. Data Organization

From the selected articles, we organized data by author(s), place of origin, publication year, study type, detected arrhythmia(s)/ECG deviation(s), sample size, average age of participants, and the smartwatches employed.

## 3. Results

We included a total of 57 articles in our research. Among the scientific works, there were 33 case reports or case series, and in addition, we selected 24 cohort studies where various arrhythmias and ECG abnormalities were recorded (Figure 1).

### 3.1. Case Reports

We selected 33 articles from case reports and case series detailing events involving a total of 44 patients. Most of the reports come from the United States (*n* = 12). In the selected articles, the youngest subject was 10 days old, and the oldest was 72 years old. In most cases (*n* = 30), the Apple Watch was used, while Samsung Galaxy Fit was used for arrhythmia registration in one patient. The smartwatch type was not precisely specified in two articles. Atrial fibrillation was recorded in 6 patient cases, and atrial flutter (AFL) was observed in 3 patients. The most frequently recorded arrhythmia was supraventricular tachycardia (SVT) (including atrioventricular re-entry tachycardia (AVRT) and atrioventricular nodal re-entry tachycardia (AVNRT), occurring a total of 13 times. Ventricular tachycardia (VT) was recorded in 7 patients. Third-degree atrioventricular (AV) block occurred 5 times. Other arrhythmias or ECG abnormalities (sinus bradycardia, sinus tachycardia, sick sinus syndrome (SSS)/tachycardia-bradycardia syndrome, Wolff–Parkinson–White (WPW) syndrome, bigeminy, ST-segment elevation/depression, ventricular fibrillation (VF) was described once each. (Table 1: Summary table of case descriptions).

### 3.2. Cohort Studies

Among the cohort studies, 24 articles met the inclusion criteria. Most of these studies originated from the United States (*n* = 6). The cardiac arrhythmia investigated most often was atrial fibrillation, documented in a total of 1294 cases throughout the studies. Other arrhythmias included atrial flutter (77 cases), atrioventricular nodal reentrant tachycardia (64 cases), atrioventricular reentrant tachycardia (36 cases), (paroxysmal) supraventricular tachycardia (PSVT) or sinus tachycardia (27 cases), ventricular tachycardia (5 cases), and second- or third-degree atrioventricular block (49 cases). Sinus bradycardia was recorded in 27 cases.

The cohort studies showed that the smartwatch that was used the most for ECG recordings was the Apple Watch, which was employed in 4479 cases. Additionally, ECGs that used Samsung (*n* = 978) were recorded 2743 times, Withings (*n* = 942), Fitbit (*n* = 360), Garmin (*n* = 223), Acer (*n* = 116), Huawei (*n* = 100), and Polar (*n* = 24) devices. (Table 2: Summary table of cohort studies).

Of the cohort studies, the first was conducted by Seshadri et al. They aimed to evaluate the precision of the Apple Watch during exercise of fifty patients with common cardiac arrhythmias like atrial fibrillation. They compared its accuracy against telemetry. The findings of this preliminary clinical investigation revealed a correlation coefficient of 0.7 between all Apple Watch readings and telemetry. Additionally, the Apple Watch exhibited greater accuracy in assessing heart rate among patients with atrial fibrillation compared to those without (rc = 0.86 for patients in AF, versus rc = 0.64 for patients not in AF) [51].

Hwang et al. conducted a study to evaluate the precision of three smartwatch models: the Apple Watch Series 2, the Samsung Galaxy Gear S3, and the Fitbit Charge 2. This research involved 51 patients with a history of paroxysmal supraventricular tachyarrhythmia (SVT) or paroxysmal palpitations. Patients were randomly assigned to wear two different devices. The initial heart rate measurements showed accuracies of 100%, 100%, and 94% for Apple, Samsung, and Fitbit, respectively. During induced SVT, in which heart rates ranged from 108 to 228 beats per minute, the accuracy was 100%, 90%, and 87% for Apple, Samsung, and Fitbit, respectively. While the devices demonstrated acceptable accuracy, it tended to decrease as heart rate increased and exhibited variations between the different models [52].

Ploux et al. evaluated the sensitivity and specificity of the Apple Watch Series 4 among 260 patients, both with and without a history of cardiovascular disease. Their findings indicate that the Apple Watch Series 4 can detect ECG abnormalities with a sensitivity of 91% and a specificity of 94% (95% CI) [53].

Sequeira et al. investigated the precision of four common wearable devices (Apple Watch, Fitbit Charge HR, Garmin VivoSmart HR, and Polar A360) in monitoring heart rate during episodes of paroxysmal supraventricular tachycardia (SVT). Their study involved 52 patients. The researchers concluded that all wearable devices showed inaccuracy for short-duration (<60 s) SVT episodes. Only the Apple Watch (23 out of 23) and Polar (19 out of 21) devices demonstrated an accuracy exceeding 90% for long-duration (≥60 s) SVT episodes [54].

Leroux et al. evaluated the sensitivity and specificity of the Apple Watch in 110 children, ranging from 1 week to 16 years old, who had either normal (*n* = 75) or abnormal (*n* = 35) 12-lead ECGs. The smartwatch tracings showed a sensitivity of 84% and specificity of 100% in detecting abnormal ECG [55].

Koshy et al. assessed the accuracy of heart rate measurement using the Apple Watch Series 1 and Fitbit Blaze among patients diagnosed with atrial fibrillation (AF) and atrial flutter (AFL). The Apple Watch demonstrated accuracies of 86%, 100%, and 99% for AF, AFL, and both conditions, respectively, when compared to an ECG monitor. Similarly, the Fitbit showed accuracies of 87%, 99%, and 98% for AF, AFL, and both conditions, respectively, when compared to an ECG monitor [56].

Abu-Alrub et al. conducted a comparison of the diagnostic capabilities for detecting atrial fibrillation (AF) among three commercially available smartwatches. Their study involved 100 patients with AF and 100 patients with sinus rhythm. They found that the Apple Watch Series 5, the Samsung Galaxy Watch Active 3, and the Withings Move ECG exhibited sensitivities/specificities of 87%/86%, 88%/81%, and 78%/80%, respectively (*p* < 0.05) [57].

Han et al. developed an algorithm aimed at detecting atrial fibrillation using a Samsung Simband 2. Their study involved 35 participants. The sensitivity, specificity, positive predictive value, negative predictive value, and accuracy for subjects with atrial fibrillation were 92%, 96%, 85%, 98%, and 95%, respectively [58].

Mannhart et al. conducted a study to evaluate the accuracy of five smart devices in detecting atrial fibrillation compared to a physician-interpreted 12-lead electrocardiogram. They prospectively analyzed 201 patients, among whom 62 had atrial fibrillation. The sensitivity and specificity for atrial fibrillation detection were similar across devices: 85% and 75% for the Apple Watch 6, 85% and 75% for the Samsung Galaxy Watch 3, 58% and 75% for the Withings Scanwatch, 66% and 79% for the Fitbit Sense, and 79% and 69% for the AliveCor KardiaMobile, respectively. In terms of patient preference, the Apple Watch ranked highest (preferred by 39% of participants) [59].

Racine et al. wanted to evaluate the precision of the Apple Watch ECG in detecting atrial fibrillation (AF) among 734 patients, of whom 21% were diagnosed with AF and had various ECG abnormalities in their study. Upon excluding unclassified ECGs from the analysis, the sensitivity was found to be 88% (95% CI 82–93%), and specificity was 98% (95% CI 97–99%). However, when unclassified ECGs were considered as false results, the sensitivity and specificity for AF detection were 69% (95% CI 61–76%) and 81% (95% CI 76–84%), respectively [60].

Pengel et al. evaluated the diagnostic precision of various ECG-based devices in comparison to the standard 12-lead ECG in a cohort of 222 patients. Their study found that for atrial fibrillation (AF) detection, the Withings Scanwatch achieved 100% accuracy, sensitivity, and specificity. Additionally, only 5% of cases were deemed uninterpretable with this smartwatch. The Kardia 6L demonstrated 97% accuracy, 100% sensitivity, and 97% specificity, albeit with 31% of cases were uninterpretable [61].

Pepplinkhuizen et al. investigated the effectiveness of the Apple Watch (AW) ECG in detecting atrial fibrillation (AF) among patients scheduled for electrical cardioversion (ECV). Their study involved obtaining AW ECGs before and after ECV, with up to three attempts made in case of unclassified recordings. Sensitivity, specificity, and kappa coefficient were calculated for analysis. A total of 65 AF and 64 sinus rhythm measurements were recorded. The initial AW measurement showed a sensitivity of 93.5% and a specificity of 100% (κ = 0.94). Subsequent measurements yielded a sensitivity of 94.6% and specificity of 100% (κ = 0.95) for the second attempt and a sensitivity of 93% and specificity of 96.5% (κ = 0.90) for the third attempt [62].

Rajakariar et al. assessed the accuracy of using an Apple Watch with AliveCor KardiaBand (KB) for diagnosing atrial fibrillation (AF) in comparison to a 12-lead ECG. The KB, when paired with a smartwatch, provided an automated diagnosis of either AF or sinus rhythm. The sensitivity and specificity of KB were 94.4% and 81.9%, respectively, with a positive predictive value of 54.8% and a negative predictive value of 98.4%. The agreement between the diagnosis from the 12-lead ECG and KB was moderate, especially when including unclassified tracings (κ = 0.60, 95% CI 0.47 to 0.72) [63].

Wasserlauf et al. recruited thirty participants for their study, aiming to evaluate the precision of the Apple Watch in detecting atrial fibrillation. Their primary goal was to ascertain the accuracy of the irregular rhythm notification (IRN) among individuals previously diagnosed with non-permanent AF. The study found no instances of false positive IRN detections, achieving a sensitivity of 72%, a specificity of 100%, a PPV (Positive predictive value.) of 100%, and a NPV (Negative predictive value) of 90% [64].

Chang et al. examined the precision of the Garmin Forerunner 945 smartwatch in identifying atrial fibrillation (AF) in comparison to a Holter electrocardiogram. Their study involved 200 participants. The sensitivity, specificity, positive predictive value, and negative predictive value for AF detection among participants were 97.3%, 88.6%, 91.6%, and 96.3%, respectively. The accuracy of the Garmin smartwatch was reported at 93.5% [65].

Wyat et al. aimed to characterize the assessments of patients who seek medical attention after detecting an abnormal pulse using the Apple Watch. They conducted a retrospective analysis of patients evaluated for an abnormal pulse detected via the Apple Watch over a four-month period. Out of the 264 patients included in the study, clinical documentation explicitly noted an abnormal pulse alert in 41 patients (15.5%). Preexisting atrial fibrillation was identified in 58 patients (22.0%). Only 30 patients (11.4%) received a clinically actionable cardiovascular diagnosis of interest, with 6 out of 41 patients (15%) who received an explicit alert among them [66].

Badertscher et al. made a prospective observational study involving patients attending a cardiology service at a tertiary referral center. Their objective was to evaluate the diagnostic accuracy of the intelligent ECG feature of the Withings Scanwatch in detecting atrial fibrillation (AF) compared to a concurrently obtained cardiologist-interpreted 12-lead ECG. In total, AF was diagnosed in 34 patients (11%). Among the ECG tracings analyzed by the algorithm, it demonstrated a sensitivity of 76% (95% CI 55–91%), a specificity of 99% (95% CI 97–100%), and a Kappa coefficient of 0.72 when compared to cardiologist-interpreted 12-lead ECGs [67].

Ford et al. conducted a comparative analysis between the Apple Watch Series 4 (AW) and the AliveCor KardiaBand (KB) for the detection of atrial fibrillation (AF) in a cohort of 125 patients. The results showed that AW automatically detected AF with an accuracy of 93%, a sensitivity of 50%, a specificity of 100%, a positive predictive value of 100%, and a negative predictive value of 92%. KB automatically detected AF with an accuracy of 94%, a sensitivity of 96%, a specificity of 93%, a positive predictive value of 84%, and a negative predictive value of 99% [68].

Lee et al. compared the Apple Watch Series 4 (AW) and KardiaMobile (KM), involving 200 participants in their study. The accuracy of rhythm detection for sinus rhythm was found to be 100% for AW and 99.03% for KM. In detecting atrial fibrillation, AW exhibited an accuracy of 90.48%, whereas KM achieved 100% accuracy. Regarding heart rate accuracy for sinus rhythm, KM showed 94.39% accuracy, while the AW photoplethysmography function had 90.65% accuracy, and the AW ECG function had 96.26% accuracy. For heart rate accuracy during atrial fibrillation, KM demonstrated 91.30% accuracy, while the AW photoplethysmography function showed 82.61% accuracy, and the AW ECG function exhibited 86.96% accuracy [69].

Roelle et al. assessed the effectiveness of digital health technologies in pediatric electrophysiology telehealth consultations. Providers evaluated the data quality from these devices using a post-visit usability survey. Regarding ECG devices, providers reported high-quality tracings from KardiaMobile (62%; 18/29), Apple Watch (93%; 28/30), and Coala monitor (86%; 24/28) [70].

Liao et al. evaluated the Acer Leap Ware smartwatch for its ability to detect atrial fibrillation (AF). Data were gathered from patients undergoing radiofrequency or cryotherapy ablation for AF. A total of 116 patients were enrolled, of which 76 had previously been diagnosed with paroxysmal AF and 40 with persistent AF. The overall accuracy of the smartwatch was summarized as 95.02%, with a sensitivity of 95.68% and specificity of 93.66% [71].

Dörr et al. utilized the photoplethysmography algorithm and discovered a sensitivity of 93.7% (95% CI: 89.8% to 96.4%), a specificity of 98.2% (95% CI: 95.8% to 99.4%), and an accuracy of 96.1% (95% CI: 94.0% to 97.5%) for detecting atrial fibrillation with a Samsung Gear Fit 2 [72].

Liu et al. employed a Huawei Watch GT 2 Pro ECG edition to identify arrhythmias in a cohort of 100 patients. Throughout their investigation, they recorded 52 instances of bradyarrhythmias, encompassing Mobitz I, Mobitz II, and third-degree atrioventricular block, as well as 16 occurrences of tachyarrhythmias, including atrial fibrillations and atrial flutters [73].

Feldman et al. aimed to provide real-world insights into the proportion of individuals who would potentially benefit from anticoagulation therapy if diagnosed with atrial fibrillation using data from wearable devices. This study utilized electronic health records (EHR) and Apple Watch data obtained from an observational cohort comprising 1802 patients. Utilizing this dataset, they estimated the number of high-risk patients eligible for anticoagulation based on their medical history, Apple Watch usage patterns, and atrial fibrillation (AF) risk determined by a validated model. Considering the characteristics of this cohort, they found that, on average, 0.25% (*n* = 4.58, 95% CI, 2.0–8.0) of patients could be considered suitable candidates for initiating anticoagulation therapy due to AF detection through their Apple Watch [74].

## 4. Discussion

Numerous studies have explored the capability of various smartwatches, including Apple Watch Series 4^®^, Samsung Simband^®^, Samsung Galaxy Watch 3^®^, Huawei Watch GT 2 Pro^®^, Fitbit Sense 2^®^, Withings Scanwatch^®^, Garmin Venu 2^®^, Polar A360^®^, Acer Leap Ware^®^, and their subsequent generations, to detect both brady- and tachyarrhythmias. Additionally, there are non-invasive devices such as AliveCor KardiaMobile^®^, ATsens^®^, Polar H10^®^, or Coala Heart Monitor^®^, and invasive measurement methods, such as Implantable Loop Recorder or Implantable Cardiac Monitor, for continuous heart rhythm monitoring.

The systematic literature review aimed to collect articles on how smartwatches were utilized for detecting arrhythmias.

The case reports and case series highlight key demographic information such as the age range of patients, the geographic distribution of cases, and the prevalence of specific arrhythmias recorded. The inclusion of patients spanning from 10 days old to 72 years old emphasizes the broad applicability of smartwatch-based arrhythmia monitoring across different age groups. Most of the cases originated from the United States, suggesting a potential concentration of research and clinical use of smartwatches for cardiac monitoring in this region. Atrial fibrillation and supraventricular tachycardia were the most recorded types of arrhythmias. This reflects the known prevalence of these arrhythmias in clinical practice and presents the importance of early detection and monitoring, particularly in high-risk populations. The smartwatch model that is used most often is the Apple Watch, which suggests its popularity and reliability.

In the cohort studies, our focus was on understanding the effectiveness of smartwatches in arrhythmia detection. The most studied arrhythmia was atrial fibrillation, with Apple Watch being the predominant device used for its detection. Studies revealed that the Apple Watch, either standalone or supplemented with KardiaBand, demonstrated over 90% accuracy in AF detection [53,63,68,69]. Diagnostic sensitivity and specificity were also consistently around 90% [53,57,60,62,63,64,68]. Moreover, the Apple Watch proved effective in accurately determining heart rate, even during tachyarrhythmias [51,52,56]. Similar results were observed with smartwatches from other manufacturers, including Samsung, Withings, Fitbit, Garmin, Huawei, and Acer [52,56,57,58,61,65,67,71,72]. Apart from AF, these devices demonstrated capability in detecting various other arrhythmias, such as second- and third-degree atrioventricular block, atrial flutter, atrial tachycardia, supraventricular tachycardia, atrioventricular nodal reentrant tachycardia, atrioventricular reentrant tachycardia, or ventricular tachycardia [52,53,54,55,60,70,73]. In addition to having good accuracy, as well as high sensitivity and specificity, smartwatches can be used easily and conveniently, which is why the majority of research participants prefer smartwatches, above all the Apple Watch, over other ECG-capable devices [59,61,70]. Their wide applicability is also facilitated by the fact that they can be used not only for adults but also for children where necessary [55,70]. Despite the convenience and accuracy of smartwatches, they are still underutilized in clinical practice for prevention and therapy adjustment. This is despite potential benefits, such as aiding in initiating anticoagulant therapy in patients with detected atrial fibrillation [74]. It is important to acknowledge false-positive events, as smartwatches may incorrectly indicate arrhythmias, potentially contributing to the burden on the healthcare system [66].

## 5. Conclusions

Our systematic literature review presents a comprehensive overview of the utilization of smartwatches for monitoring cardiac arrhythmias, focusing on both case reports and cohort studies. Let us delve into some key points drawn from these findings:Diversity in patient demographics and arrhythmias: The study encompassed a wide range of patients, spanning from a 1-week-old infant to a 91-year-old individual, showcasing the applicability of smartwatch technology across various age groups. Moreover, the diversity of recorded arrhythmias, including atrial fibrillation, atrial flutter, supraventricular tachycardia, ventricular tachycardia, and others, highlights the versatility of smartwatches in detecting different cardiac anomalies.Prevalence of Apple Watch usage: The Apple Watch emerged as the most utilized and most reliable smartwatch for arrhythmia monitoring in both case reports and cohort studies. This prevalence might be attributed to its widespread availability, user-friendly interface, and integration with healthcare systems.Accuracy and precision across different studies: Several cohort studies evaluated the accuracy of smartwatch models in detecting cardiac arrhythmias. Findings varied across studies, with some reporting high sensitivity and specificity, particularly for atrial fibrillation detection, while others noted variations in accuracy depending on the smartwatch model and type of arrhythmia.Comparison studies: Comparative studies, such as those assessing different smartwatch models or comparing smartwatch performance with standard ECG monitoring, provided valuable insights into the strengths and limitations of each device. These comparisons aid in guiding clinicians and patients in selecting the most suitable device for their specific monitoring needs.Clinical implications: The study’s findings have significant clinical implications, particularly in the early detection and management of cardiac arrhythmias. Smartwatches offer the potential for continuous monitoring outside clinical settings. It could be helpful for monitoring a patient in need or who underwent a major intervention, to improve the patient’s outcome.Challenges and future directions: Despite promising results, challenges such as accuracy during high heart rates and variability across different smartwatch models underscore the need for further research and technological advancements. Future studies may focus on enhancing the accuracy, reliability, and usability of smartwatch-based arrhythmia detection systems.

In summary, this study provides insights into the evolving role of smartwatches in cardiac arrhythmia monitoring. While advancements in wearable technology hold promise for revolutionizing healthcare delivery, continued research and validation are essential to optimize their clinical utility and ensure patient safety and efficacy.

## Figures and Tables

**Figure 1 healthcare-12-00892-f001:**
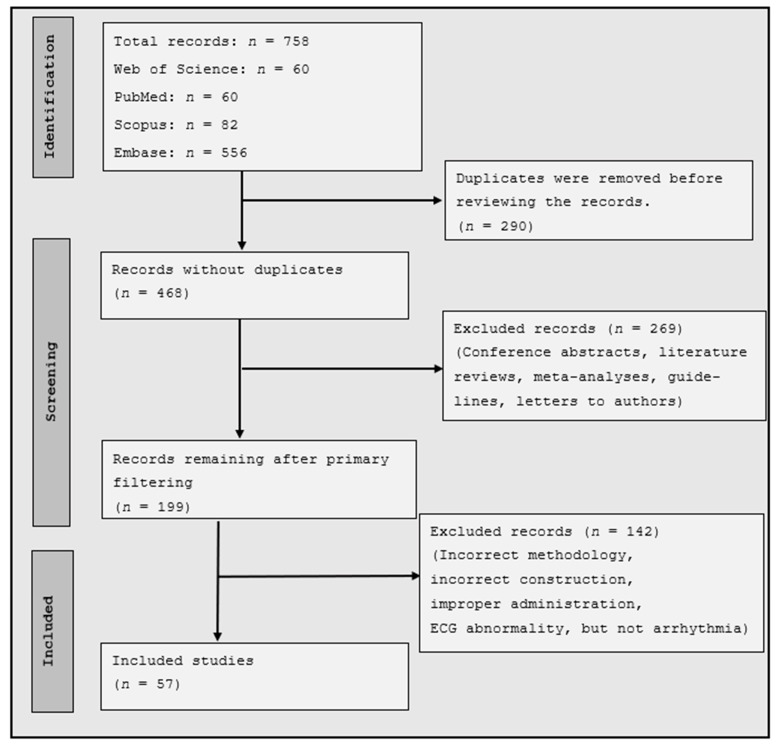
PRISMA Flow Diagram.

**Table 1 healthcare-12-00892-t001:** Summary table of case descriptions.

Authors	Country	Year	Arrhythmia Type/ECG Abnormality	Number of Patients	Age	Smartwatch Type
Sanchez et al. [18]	USA	2022	Sinus bradycardia	1	32	NA
Kasai et al. [19]	Japan	2021	AVNRT, AVRT	1	52	Apple Watch
Ocher et al. [20]	USA	2023	VT	1	36	Apple Watch
Hawrysko et al. [21]	Poland	2022	AVNRT	1	35	Apple Watch
Al-Sudani et al. [22]	USA	2023	Third-degree AV block	1	44	Apple Watch
Siddeek et al. [23]	USA	2020	AVNRT	1	16	Apple Watch
Wu et al. [24]	Taiwan	2022	SVT	3	59, 60, 48	Apple Watch
Leroux et al. [25]	France	2021	Sinus tachycardia, SVT, Third-degree AV block	3	10 days, 4 months, 16 months	Apple Watch
Kassam et al. [26]	Tanzania	2021	AVRT	1	42	Apple Watch
Ringwald et al. [27]	Switzerland	2020	VT	1	45	Apple Watch
Goldstein and Wells [28]	South Africa	2019	AFL	1	56	Apple Watch
Bedi et al. [29]	USA	2023	AF	1	25	NA
Bogossian et al. [30]	Germany	2020	AVNRT	1	65	Apple Watch
Gu et al. [31]	Canada	2022	VT	1	64	Apple Watch
Burke et al. [32]	USA	2020	VT	2	60, 63	Apple Watch
Jeong [33]	South Korea	2022	AVNRT	1	23	Apple Watch
Ahmed et al. [34]	USA	2020	AFL	1	54	Apple Watch
Yeo et al. [35]	Singapore	2021	SVT with aberrant conduction	1	NA	Apple Watch
Russo et al. [36]	Italy	2023	“Narrow-wide-narrow” QRS tachycardia	1	NA	Apple Watch
Glöckner et al. [37]	Germany	2022	NSVT, ST-segment elevation	1	44	Apple Watch
Leroux et al. [38]	France	2022	SVT, WPW syndrome, Third-degree AV block	6	5, 6, 7, 9, 11, 13	Apple Watch
Overbeek et al. [39]	USA	2019	Third-degree AV block	1	60	Apple Watch
Yerasi et al. [40]	USA	2020	Third-degree AV block	1	68	Apple Watch
Itoh [41]	Japan	2022	AF	1	60	Apple Watch
Mun et al. [42]	South Korea	2021	WPW syndrome	1	26	Samsung Galaxy Fit
Delinière et al. [43]	Switzerland	2021	ST-segment depression, VT	1	45	Apple Watch
Walker et al. [44]	USA	2023	AFL	1	37	Apple Watch
Weichert [45]	UK	2019	AF	1	59	Apple Watch
Samal et al. [46]	USA	2020	AF	1	39	Apple Watch
Jariwala and Jadhav [47]	India	2021	AF, SSS (tachycardia-bradycardia syndrome)	2	72, 69	Apple Watch
Pasli and Imamoglu [48]	Turkey	2023	Bigeminy	1	41	Apple Watch
Patel and Tarakji [49]	USA	2021	AF	1	70	Apple Watch
Provencio and Gil [50]	Spain	2022	ST-segment depression, PVCs, VF	1	72	Apple Watch

NA = Not available.

**Table 2 healthcare-12-00892-t002:** Summary table of cohort studies.

Author(s)	Country	Year	Arrhythmia Type/ECG Abnormality	Number of Patients	Average Age	Smart-Watch Type(s)
Seshadri et al. [51]	USA	2019	AF	50	61.4 ± 10.4	Apple Watch
Hwang et al. [52]	South Korea	2019	PSVT	51	44.4 ± 16.6	Apple Watch, Samsung Galaxy Gear, Fitbit Charge
Ploux et al. [53]	France	2022	Sinus bradycardia, Second-, Third-degree AV block, AF, AFL/AT, ST-, T-wave changes, RBBB, LBBB, Pathological Q-wave	260	66 ± 6	Apple Watch
Sequeira et al. [54]	Canada	2020	AVNRT, AVRT	52	52.3 ± 17.2	Apple Watch, Fitbit Charge, Garmin Vivo-Smart, Polar A360
Leroux et al. [55]	France	2022	BBB, AV block, WPW, SVT, Long-QT	110	1 week–16 years	Apple Watch
Koshy et al. [56]	Australia	2018	AF, AFL	102	68 ± 15	Apple Watch, Fitbit Blaze
Abu-Alrub et al. [57]	France	2022	AF	200	62 ± 7	Apple Watch, Samsung Galaxy Watch, Withings Move
Han et al. [58]	USA	2021	AF	35	50–91	Samsung
Mannhart et al. [59]	Switzerland	2023	AF	201	66.7	Apple Watch, Samsung Galaxy Watch, Withings Scan-watch, Fitbit Sense, AliveCor Kardia-Mobile
Racine et al. [60]	Canada	2022	AF, AFL/AT, VT, SVT, sinus dysfunction, second-and third-degree AV block, ventricular ectopic beats, RBBB, LBBB	734	66	Apple Watch
Pengel et al. [61]	Netherlands	2022	AF	222	40 ± 17	Withings Scan-watch
Pepplinkhuizen et al. [62]	Netherlands	2022	AF	74	67.1 ± 12.3	Apple Watch
Rajakariar et al. [63]	Australia	2020	AF	200	67 ± 16	Apple Watch
Wasserlauf et al. [64]	USA	2023	AF	30	65.4 ± 12.2	Apple Watch
Chang et al. [65]	Taiwan	2022	AF	200	66.1 ± 12.6	Garmin
Wyatt et al. [66]	USA	2020	AF	264	55	Apple Watch
Badertscher et al. [67]	Switzerland	2022	AF	319	67	Withings Scan-watch
Ford et al. [68]	Australia	2022	AF	125	76 ± 7	Apple Watch
Lee et al. [69]	Canada	2022	AF	200	65.6 ± 14.6	Apple Watch
Roelle et al. [70]	USA	2022	SVT, Arrhythmia Syndrome, Syncope, Sinus arrest, Sinus tachycardia	30	11.6	Apple Watch
Liao et al. [71]	Taiwan	2022	AF	116	59.6 ± 11.4	Acer Leap Ware
Dörr et al. [72]	Germany	2019	Paroxysmal Fibrillation (PF)	508	76.4	Samsung
Liu et al. [73]	China	2022	Brady-arrhythmia, Mobitz I, Mobitz II, Third-degree AV block, LBBB, Tachy-arrhythmia, AF, AFL	100	73.1 ± 7.6	Huawei
Feldman et al. [74]	USA	2022	Paroxysmal Fibrillation (PF)	1802	45.96	Apple Watch

## Data Availability

Not applicable.

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
