# Peer review of "Detection of Arrhythmias Using Smartwatches—A Systematic Literature Review"

_healthcare, 2024, doi:10.3390/healthcare12090892_

Round 1
Reviewer 1 Report
Comments and Suggestions for Authors
The study performs a systematic review of articles and reports on the usage of smartwatch to detecte Arrhythmias. The authors, however only sought for articles and reports published till August 2023. It will be jusdicious to include studies publishe after August 2023. This could support to answer question raised in the conclusion.
Despite that the study make substantial contribution to the literature, the authors fail to present the paper in the good shape as requested by the 2022 prisma framework.
The discussion shows also some weeaknesses. I cannot figure out any discussion on the strenght of each device in detecting the disease. The authors are well advised to add this discussion or this mention taht studies included did not discussion the accuracy level and other specific aspects.
Reviewer 2 Report
Comments and Suggestions for Authors|
Does the introduction provide sufficient background and include all |
YES |
|
|
Are all the cited references relevant to the research? |
YES |
|
|
Is the research design appropriate? |
YES |
|
|
Are the methods adequately described? |
YES |
|
|
Are the results clearly presented? |
YES |
|
|
Are the conclusions supported by the results? |
Can be improved |
The conclusion accurately reflects the findings but could be strengthened by emphasizing the need for future research on broader integration into healthcare and data reliability. |
|
Lingua Inglese |
Moderate editing of English language required
|
Overall, the English of the text is good, but it could be improved with some revisions to make it more transparent, precise, and professional. Here are some examples:
|
|
Originality / Novelty |
Medium |
The text presents an updated literature review on the use of smartwatches for the detection of cardiac arrhythmias. The topic is of current interest and constantly evolving, with new studies published regularly. However, the text does not present any real novelty in terms of results or conclusions. |
|
Significance of Content |
High |
The text provides a comprehensive and well-documented overview of the potential of smartwatches for the detection of cardiac arrhythmias. The information presented is accurate and up-to-date, and the text is well supported by bibliographic references. |
|
Quality of Presentation |
Medium |
The text is generally clear and well written. However, some sentences could be rephrased to make them more concise and fluent. In addition, the formatting of the text could be improved to make it more enjoyable to read. |
|
Scientific Soundness |
High |
The text is based on a solid review of the scientific literature. The information presented is accurate and supported by scientific data. The text is also well referenced, with links to relevant studies and articles. |
|
Interest to the readers |
High |
The text is of sure interest to an audience of cardiologists, electrophysiologists, general practitioners and other healthcare professionals interested in the use of innovative technologies for the diagnosis and management of cardiac arrhythmias. The text could also be interesting for a wider audience of people interested in health and well-being. |
|
Overall Merit |
Medium |
The text is a good example of a literature review. It is well written, informative and accurate. However, some minor revisions could improve its clarity, fluency and formatting. |
Comments:
Lines 71-90: The section 2.1 “Procedure for Literature Search”, could be more detailed regarding the inclusion criteria used to select studies. It would be useful to specify, for example, The time period considered for the search (e.g., from 2010 to 2023); the types of studies considered (e.g., randomized controlled trials, observational studies, cohort studies, case-control studies); the languages of the studies considered and the keywords and Boolean operators used for the search (e.g.,("smartwatch" OR "smart watch") AND ("cardiac arrhythmia" OR "atrial fibrillation" OR ecc.).
Lines 91-93: The section 2.2 “Quality Assessment”, It could be useful to include an assessment of the risk of bias of the studies included in the review through a table or graph.
Comments on the Quality of English Language
Overall, the English of the text is good, but it could be improved with some revisions to make it more transparent, precise, and professional.
Here are some examples:
- Some sentences are a bit long and complex and could be broken down into shorter sentences.
Some words could be replaced with more precise or appropriate terms
Reviewer 3 Report
Comments and Suggestions for Authors
The paper presents A Systematic Literature Review on the use of smartwatches for the detection of Arrhythmias. Different types of studies have been covered. Some points need to be addressed to further improve the review.
In Table 1, the column "study type is repeated for each entry. Only one entry is found with the type of image recongination (I think there is a spelling mistake) whereas the rest are case reports. This can be mentioned in the table caption or footnote. A similar issue also exists in Table 2. The reference to the table in the text is incorrect.
The articles included in Table 2 are discussed in detail. however, no such discussion is available for articles included in Table 1. Including the details will be helpful for readers.
The discussion is too generic and does not convey any findings by analyzing the data collected. The authors need to include a discussion that satisfies the main objective of the review. This will help in justifying the statement in the abstract "Through further research, it may be possible to develop a care protocol that integrates arrhythmias recorded by smartwatches, allowing for timely access to appropriate medical care for patients.". The potential problems in the existing protocols need to be identified.
There is an error of reference in line 131. The paper needs to be proofread for typos and spelling mistakes.
Comments on the Quality of English LanguageWritten in comments.
Round 2
Reviewer 1 Report
Comments and Suggestions for Authors
This paper is reporting a systematic literatur review on detecting Arrhythmias using smartwatches.
The authors are predenting having performed a systematic literature review and have included case reports published between and Jan. 2029 August 2023.
They have sought only 4 academic data bases. This limits the scope of the search. For example, they did not include an article publishe in March 2023 which performed similar work (see below).
It seems like the authors paraphrase the below cited study.
Anatol J Cardiol 2023 Mar;27(3):126-131. doi: 10.14744/AnatolJCardiol.2023.2799. Arrhythmias Beyond Atrial Fibrillation Detection Using Smartwatches: A Systematic Review

minor
Reviewer 3 Report
Comments and Suggestions for Authors
Most of the comments are addressed. Some readability issues still exist e.g., the caption of table 2 is “summary of the cohort studies” then including a column with the name study type is useless. All types are cohort which is mentioned in the caption. Similarly issue also exist in table 1.
